# Risk and Protective Factors Experienced by Fathers of Refugee Background during the Early Years of Parenting: A Qualitative Study

**DOI:** 10.3390/ijerph19116940

**Published:** 2022-06-06

**Authors:** Eleanor Bulford, Alison Fogarty, Rebecca Giallo, Stephanie Brown, Josef Szwarc, Elisha Riggs

**Affiliations:** 1Intergenerational Health, Murdoch Children’s Research Institute, Melbourne 3052, Australia; ebulford@student.unimelb.edu.au (E.B.); ali.fogarty@mcri.edu.au (A.F.); rebecca.giallo@mcri.edu.au (R.G.); stephanie.brown@mcri.edu.au (S.B.); 2Department of Paediatrics, The University of Melbourne, Melbourne 3052, Australia; 3Centre for Social and Early Emotional Development, School of Psychology, Faculty of Health, Deakin University, Geelong 3216, Australia; 4Department of General Practice, The University of Melbourne, Melbourne 3052, Australia; 5South Australian Health and Medical Research Institute, Adelaide 5001, Australia; 6Victorian Foundation for Survivors of Torture Inc., Brunswick 3056, Australia; szwarcj@foundationhouse.org.au

**Keywords:** fathers health, refugee health, cross-cultural research, qualitative study, socioecological model

## Abstract

Fathers of refugee background with young children can experience significant mental health difficulties, with the potential for intergenerational impacts. This study aimed to explore how fathers of refugee background experience risk and protective factors for their own health and wellbeing during the early years of parenting. Semi-structured interviews and one semi-structured focus group were conducted with fathers of refugee background, with young children (0–5 years), who had settled in Australia. Transcribed interviews were analysed using thematic analysis, informed by the socioecological model of health. A total of 21 fathers participated in the study. Risk factors experienced included: prior experiences of trauma, reduced access to family support in Australia, adjustments in parenting roles, and the challenges of learning a new language and securing employment. Fathers drew on a number of sources of strength, including a sense of joy from fatherhood and support from partners, families, and communities. While most fathers regularly accompanied their partners and children to healthcare appointments, they were rarely asked by healthcare professionals about their own needs. Our findings support the idea that there is a need for greater assistance for fathers, particularly for navigating issues arising from the settlement process. Healthcare services working with families of refugee background must adopt a father-inclusive, trauma-informed approach that is responsive to fathers’ needs.

## 1. Introduction

It is well established that early childhood is a vital developmental stage [1]. Childhood development theory has long recognised that healthy development in the early years depends on caregivers’ ability to provide care to young children that is nurturing and responsive [2]—a task that is closely contingent on the caregiver’s own mental and physical wellbeing. The early years of parenting, however, can be immensely challenging. Fathers of young children are known to experience high rates of depression and anxiety [3,4,5]. In Australia, fathers of children under five are at increased risk of mental health difficulties, with approximately one in ten experiencing clinically significant levels of psychological distress [6]. Poor paternal mental health is associated with adverse emotional, behavioural, and developmental outcomes in young children [7,8].

For fathers of refugee background, the challenges of parenting young children may be experienced in the context of a complex psychosocial history, including experiences of multiple traumatic events across the refugee journey and major adjustments following settlement in a new country [9,10]. Acculturation following settlement, defined as the processes of cultural and psychological change that arise from contact between two or more cultural groups [11], is also a recognised source of stress [12]. A high prevalence of mental health difficulties is consistently reported among people of refugee background [13,14,15]. In Australia, the prevalence of post-traumatic stress disorder (PTSD) among settled refugee populations is estimated to be between 22% and 48% [16,17]. Longitudinal data indicates that increased levels of psychological distress may persist for many years after settlement [18].

Research into the health and wellbeing of fathers of refugee background is limited. Secondary analysis of an Australian population-based cohort found that compared to fathers born in Australia, fathers of refugee background had approximately three times the odds of experiencing psychological distress during their children’s early years [19] and reported poorer overall mental and general health. This is consistent with the findings of other studies internationally, which have shown high rates of psychological distress, post-traumatic stress, anxiety, and depressive symptoms among fathers of refugee background [20,21,22,23]. Consistent with a deepening understanding of intergenerational trauma, poorer mental health among fathers of refugee background has been associated with poorer quality of father–child interactions and adverse emotional and behavioural outcomes for children [21,22,24].

One of the few qualitative studies exploring refugee background fathers’ health and wellbeing interviewed Afghan men of refugee background about their experiences of having a baby in Australia [25]. Despite playing a major role in supporting their wives and new-born infants, men taking part in this study were rarely asked by health professionals about their own health and wellbeing. Interviews with healthcare professionals indicated that they were uncertain about how to support fathers of refugee background [25]. Qualitative studies from Australia [26] and Canada [27] have examined refugee background men’s experiences of parenting after settlement, finding that many experienced differing cultural values and expectations of fatherhood in their new country. Challenges impacting upon their role as a father such as underemployment, lack of social support, and experiences of racism were also described [26,27]. Qualitative studies from Canada and Sweden that included both mothers and fathers have described how challenges of settlement such as securing housing and employment and learning a new language led to stress and uncertainty for refugee parents, as well as experiences of loneliness and isolation [28,29]. In general, however, fathers are underrepresented in research exploring the experiences of parents of refugee background [30].

It is clear from existing literature that many fathers of refugee background are likely to face immense challenges, and that the health and wellbeing of this population has potentially significant impacts upon the health of their children and families. There is, however, a lack of qualitative literature exploring fathers’ experiences of their own health and wellbeing following settlement. The aim of the current study is to explore the ways in which fathers of refugee background settling in Australia experience risk and protective factors for their own health and wellbeing during the early years of parenting. The study draws on the socioecological model of health first conceptualised by Urie Bronfenbrenner in the 1970s [31]. The socioecological model recognises the ways in which the health of an individual is determined by its interactions with multilevel factors in the immediate and more distant physical, social, and political environments [31]. More recently, Dahlgren and Whitehead’s work on social determinants of health drew upon the socioecological model to conceptualise the health of an individual as being impacted by individual lifestyle factors, social and community networks, and general socio-economic, cultural and environmental conditions [32]. Drawing on a socioecological lens to achieve a deeper understanding of the factors influencing the wellbeing of fathers of refugee background is important not only for addressing the health inequities experienced by this population, but also for optimising the health of their children and families. This study seeks to address this gap through capturing the lived experiences of fathers of refugee background with young children settled in Australia.

## 2. Materials and Methods

### 2.1. Study Design

This project is an exploratory qualitative study conducted by the Intergenerational Health Research Group, Murdoch Children’s Research Institute (MCRI) in collaboration with the Victorian Foundation for Survivors of Torture (known as Foundation House), a specialist refugee trauma agency [33]. The study was conducted in Melbourne, Australia, in 2017. The project design drew on community engagement and participatory research methods, using a framework previously developed by the research group [34], and engaged bicultural Community Liaison Workers to support recruitment and data collection.

### 2.2. Study Population and Recruitment

Prospective participants who were eligible to participate in the study were men of refugee background, aged 18 years and older, with children aged 0–5 years old, living in Melbourne, Australia. We took a purposive and snowball approach to sampling, with recruitment occurring in the community with the assistance of bicultural Community Liaison Workers employed by Foundation House. Community Liaison Workers approached potential participants and once recruited, each participant was asked if they knew of other fathers who may like to participate. Community Liaison Workers explained the project in detail in participants’ preferred language and informed verbal or written consent to participate was obtained, depending on the participant’s literacy needs.

### 2.3. Data Collection

ER conducted semi-structured individual interviews and one semi-structured focus group together with a bicultural Community Liaison Worker, who also provided interpretation in cases where the participant’s preferred language was not English. Some participants preferred to be interviewed without a Community Liaison Worker, as their English was sufficient, and they didn’t feel the need for additional support. The interview questions centred around experiences of being a parent in a new country, experiences of health, challenges faced, sources of support, interactions with healthcare services, the impact of being a father on health and wellbeing, and perceived needs for support. Participants received a $30 supermarket gift voucher to thank them for participating. All interviews and the focus group were audio recorded using a digital recorder and subsequently confidentially transcribed by a professional transcription service providing a transcript in English. EB cross-checked all transcripts. All recordings and transcripts were stored securely in a password protected shared drive which only the research team have access to. NVivo Version 12 [35] was used to manage the data.

### 2.4. Data Analysis

We analysed transcripts using thematic analysis, as described by Green et al. [36]. Following multiple reviews of the data, we applied codes, or descriptive labels, to sections of the transcript. After consideration of the codes that had been identified, those that shared a relationship were linked to create categories, and each identified category was analysed through the lens of the research question. This process was informed by the socio-ecological model of health [31], which helped to guide the interpretation of the data into coherent themes that allowed a meaningful analysis of refugee background fathers’ experiences.

### 2.5. Ethics

Ethics approval was obtained through the Royal Children’s Hospital Human Research Ethics Committee (approval number 36192A). We drew upon principles of ethical cross-cultural research throughout the research process [37]. A trauma-informed protocol for working with refugee background communities, previously developed by the research team and Foundation House, was in place for the research team to minimise the potential for distress to arise and to respond should a participant become distressed.

## 3. Results

The final sample consisted of 21 participants. Participants’ countries of origin were Syria (identifying as Assyrian Chaldean) (6), Afghanistan (4), Burma (identifying as Karen) (4), Sierra Leone (4), and Sri Lanka (identifying as Tamil) (3). The mean age of participants was 40 years (range 28–56 years; standard deviation 6.43). Mean time in Australia was 5.5 years (range 1–15 years; standard deviation 4.48) and participants had an average of 3.1 children (range 1–6 children; standard deviation 1.55). Participants were asked to rate their English language ability as very good (seven participants), good (seven participants), OK (one participant), not so good (four participants), or no English (two participants). A total of seven participants were employed on a full-time basis, six were employed on a part-time basis, one was employed casually, and seven were not employed.

Drawing upon the socioecological framework, key themes that arose from data analysis are mapped in Figure 1. Themes are displayed as both risk (lower aspect of map) and protective (upper aspect of map) factors across the domains of the individual, social and community networks, and policies and institutions. Several important themes (displayed enclosed in rectangles) arose across more than one domain. Some themes are displayed midway between risk and protective factors, indicating the ways in which these issues encompassed the potential for both risk and protective effects.

### 3.1. Individual Factors

#### 3.1.1. Joy of Fatherhood

Many fathers described deriving a strong sense of meaning from fatherhood, and their pride in being able to fulfil their parental roles. Most fathers expressed that spending time with their children and watching them grow up provided them with a great deal of happiness.


*“What I love about it, I think just the beauty of spending time with my son and hearing what he says. The person that he is, and as a two-year-old he’s got his own character, his own personality.”*
Afghan father.

Most fathers described taking active roles in looking after their children and being involved with many different aspects of their care, including preparing food, bathing, and taking their children to school and childcare. For several, time with their children was described as a major source of strength amidst other challenges they may be facing:


*“And, sometimes when you stressful, you worry, you come home and you play with the children, all the stress is gone. It’s like a medicine, you know? It works.”*
Afghan father.

#### 3.1.2. Work/Life Balance

Fathers described having busy lives, with many managing multiple demands across work, study, and family life. Managing multiple commitments was noted by one father to impact his ability to learn English and secure a job:


*“… as fathers, we have a lot of obligations. We have to take our children and our wives to the appointment, to the doctors, to schools. We don’t have that time, enough time to go to school and learn the language”*
Assyrian father.

For many fathers, their multiple commitments resulted in limited time for themselves. Upon questioning about their personal hobbies, most participants responded that they did not have the time to spend on recreational activities such as sport or music. One father reflected on how his journey as a refugee had influenced his experience of juggling multiple commitments upon settlement:


*“As a refugee, if you want to be successful, you really need to compensate for the things that you’ve lost. So, the few years that I spent in detention and the time that I lost in Afghanistan as I couldn’t continue with my education, and then first four or five years of life here, I had to work in factory jobs to keep up supporting a big family back at home and also myself here. So, with all of those, I had to work really hard.”*
Afghan father.

The same father made the following observation about the challenges other fathers in his community had faced:


*“I could see that a lot of the Afghani fathers, especially coming from asylum seeker, refugee backgrounds, had issues… a lot of them would focus so much to get financially by that they would work, you know, 60 h, and wouldn’t be there for the kids. And, I think I really could see the need for services to communicate to these people that, you know, your ultimate goal is a better future for your children, isn’t that? And, they would be, “Yes. Absolutely.” “And, how do you see achieving that, is by working 60 h?”*
Afghan father.

#### 3.1.3. Personal Coping Mechanisms

Despite generally leading busy lives and having limited time for themselves, several fathers identified personal strategies for managing stress and looking after their own wellbeing. These included reading, listening to music, watching TV or a movie, meditation, and gardening.


*“If I notice, for instance, that becoming somehow depressed about something, one way is to take my book, play my music—I’ve got some very special songs.”*
Sierra Leonean father.

#### 3.1.4. Prior Experiences of Trauma

Several fathers described or referenced traumatic experiences across the refugee journey. These included exposure to persecution and hardship in their countries of origin and throughout the journey to Australia. For several this included time in refugee camps and immigration detention and prolonged periods of separation from their families. Their comments highlighted that the challenges of parenting in a new country were experienced in the context of a history of exposure to trauma. One father described a sense of wanting to hide his traumatic experiences from his children:


*“As a father in the camp in Burma, we have a lot of trauma experience and hardship going through in our life. It’s good not to share our trauma and hardship to the children because the children grow up in this country, better grow in the future…although we have very painful experience, we better not show the children about our painful experience, only encourage the children for the better future.”*
Karen father.

#### 3.1.5. Changes to Parental Roles

While some participants felt that their roles as fathers had not changed significantly since arrival in Australia, others described taking on a more active parenting role than what would have been expected in their country of origin. Several fathers commented that having limited family support available to them in Australia had meant they needed to take on different responsibilities in caring for their children, and others described cultural differences between Australia and their country of origin in the expectations of fathers’ roles in raising their children. Observations of other changes to gender roles, such as women taking on paid employment, were also described.


*“I can say it’s shame in Syria as a dad to take care of a child, changing her or him…the nappies, yeah… but here, I can see that there’s no difference or it’s okay for the dad to take care of the baby.”*
Assyrian father.

While several fathers commented that they felt this potential adjustment in the paternal role may pose a challenge for other fathers of refugee background, many participants in the study described embracing fatherhood and their child-rearing role in the Australian context.


*“It’s good in the new country, you can better learn the new way of nurturing the children, because you can’t do it all in the traditional way because the children grew up in a new environment, so the father needs to learn the new way.”*
Karen father.

### 3.2. Social and Community Network Factors

#### 3.2.1. Support from Close Relationships: Partner, Family, Community Members

A strong sense of teamwork with their partner was identified by many fathers as a source of strength as they raised their children and navigated the challenges of settlement together. Several described their partners as being an important source of emotional support when managing stress or uncertainty.


*“After having the first child the relationship is strengthened more and more. Everything—I think it’s love is increased and we link to each other more and our relationship is turned to like iron relationship.”*
Assyrian father.

For some fathers, members of their extended families had also settled in Australia and were described as an important part of their daily lives and source of support. For others, despite their extended families remaining overseas, these fathers described keeping in close contact with their parents, parents-in-law, and siblings and looking to these relatives for help and advice with parenting.


*“Especially with the kids, when the kids have some issue, we ask our parents, especially my mum, yeah, we have this situation, how to cope with this one, what to do.”*
Afghan father.

Most fathers described having friendships and spending time with community members in informal social situations, such as barbecues, and at religious gatherings. Some fathers regarded other community members as important sources of both support and information for parenting. Several stated that they thought greater opportunities for fathers in their communities to meet with other fathers and share experiences would be valuable.


*“It is helpful because I think every parent has its own sort of experience…when you do talk with them, also it’s a good opportunity for exchange experience, perhaps the way they do things, the way they solve the problem, I think it’s good to have that sort of conversation.”*
Afghan father.


*“This is how it works here, because if it comes from us that have lived here for long, telling our African brothers, they would tend to believe us more. Because, we have gone through and we should be able to share our experiences with them.”*
Sierra Leonean father.

However, a few fathers stated they were less likely to ask friends or community members for advice or emotional support if they were experiencing challenges.


*“So, for example, there are sometimes I might not tell them details about how I manage something, you know, some difficulty. Because, I just feel vulnerable in some ways, you know? I don’t want to be judged too much.”*
Afghan father.

#### 3.2.2. Reduced Family and Community Support in New Country

Despite the importance of support from their close relationships, several fathers reflected on the differences in family and community support structures that would have been available to them in their countries of origin, compared with Australia. This identified a gap in access to both practical and emotional support as they navigated their new lives in Australia as parents.


*“From my experience, becoming a parent for the very first time in the western world, it’s very hard, compared to in Africa. In Africa you have so many help.”*
Sierra Leonean father.

### 3.3. Policy and Institutional Level Factors

#### 3.3.1. Employment and Financial Pressures

Fathers described a sense of responsibility to provide for their families financially. This was a significant source of stress.


*“Basically, as the father, I feel the pressure of… I need to make sure that we are financially okay.”*
Afghan father.

Language and literacy barriers were described as major obstacles to being aware of job opportunities and applying for and securing employment. Other barriers cited included visa requirements and being told they did not have sufficient experience for a role.


*“Yeah, it’s very stressful looking for the job, because of the language problem and also—don’t know how to use the online job applications and online sites”*
Karen father.

#### 3.3.2. Language

The challenge of needing to learn English was an overarching issue for many fathers that impacted upon many facets of their lives. While several fathers were proficient in English prior to arriving in Australia, for others, language factors were a significant barrier for securing employment, establishing friendships in their new community, and being aware of support services available. A few fathers commented that they felt the support to learn English upon settlement had been inadequate.


*“Without having language it’s very difficult to find a job and to find employment. We feel that they pressure, pressuring…It’s during a small period of time and they’re asking us to find a job. We are not able just to say ‘Hi’ in English. How can we find a job without having this language?”*
Assyrian father.


*“If you can’t communicate, you cannot appropriately address your needs… main barrier would be language, otherwise if you know the language, then you know about the services available.”*
Afghan father.

Fathers who did not speak English well described mixed experiences of access to interpreters in healthcare settings. Several participants who required interpreters reported that they did have access to one whenever needed, while others described using family members or friends as interpreters.


*“For the first appointment I did not have an interpreter. But I asked for the telephone interpreter. And, then the next follow up appointment no interpreter. So I accompany with a friend or relative to go interpreter for me.”*
Karen father.

#### 3.3.3. Awareness of Services

Several fathers commented that a lack of awareness about the healthcare and community services available was a significant barrier to access within their community and that accessing information about services could be challenging. Language and literacy were both cited as barriers to being aware of available supports.


*“The other family told us that when we talked about a stressful time, and they told, “Why you didn’t use this service?” And we told that we didn’t know about that one.”*
Afghan father.

#### 3.3.4. Helpful Interactions with Healthcare Services

Although a lack of awareness of services was noted as a barrier, most fathers described healthcare services as being an important source of help and advice, particularly regarding the health of their children. Services cited as helpful included the maternal and child health nurse, the general practitioner (GP), and hospital maternity services. Several participants described having a great deal of confidence in their healthcare professionals and positive experiences of having received advice and support.


*“Thanks to our new home, Australia, because we have one of the best health systems in the world. I’m very proud of that. And we did get good advice from our doctors and GPs.”*
Afghan father.

#### 3.3.5. Limited Involvement of Fathers by Service Providers

Most fathers reported being in good health and did not regularly attend healthcare services for themselves. However, most fathers frequently accompanied their wives and children to maternal child health nurse or GP appointments. The majority of fathers reported that they were rarely included in these consultations or asked about their own wellbeing. A few fathers commented that their participation in this study was the first time they had been asked about their own health and wellbeing.


*“…most times I’ll be there just observing. They’ll be talking. Actually, they’ll mostly be asking my wife questions, she’ll be answering, and I’ll be just there.”*
Sierra Leonean father.


*“They weigh the baby and they check some movements and that’s all basically about it…I think this maternal child health should be more than a GP role to see the wellbeing of the mother, wellbeing of the baby, and the whole family—the father, mother, parents and the baby”*
Tamil father.

## 4. Discussion

This study sheds light on the experiences of fathers of refugee background during their children’s early years and the risk and protective factors impacting upon their wellbeing. Our findings highlighted challenges faced by fathers of refugee background arising from the processes of settling in a new country, navigating new systems, and their experiences of insufficient support throughout these processes. The challenges of learning a new language affected many aspects of fathers’ lives and the ways in which they were able to engage with their new society. Barriers to securing employment and resulting financial pressures were described as leading sources of stress for many fathers interviewed. These findings are consistent with previous research into the experiences of men of refugee background, both in the Australian and international contexts [27,38]. Our study highlights how being a parent of young children adds an additional layer of complexity to these challenges. Fathers acutely felt the responsibility to support their families financially, while juggling the demands of working or looking for work, learning a new language, and caring for their children. Further compounding this was the limited family and community support available to fathers, with most extended families and pre-existing support networks remaining overseas. This manifested in most fathers having very busy lives, with limited time for themselves. Findings highlight the impacts of refugee settlement policies upon health and wellbeing and the importance of services tailoring support to fathers’ needs and circumstances.

While there was some diversity of experience across the sample, several fathers described differences in cultural expectations of fatherhood in Australia and other changes in gender roles within families. Psychological stress arising from acculturation has been well described among refugee populations [12,26,39], along with the potential for this to contribute to changes in family dynamics and tensions within relationships [12]. A few fathers in the study commented that they had observed these issues arising within other families in their communities. Our findings indicate that fathers of refugee background may be experiencing changes in the roles they fulfil within their families. The potential impacts of this upon both mental health and family dynamics should be carefully considered by services working with families of refugee background and may be considered in future research.

Unsurprisingly, our findings indicate that many fathers of refugee background have experienced significant trauma in their countries of origin and across their journeys as refugees. Such adverse experiences have well documented links to a number of poor physical and mental health outcomes [40,41]. For fathers of refugee background, the emerging evidence of links between parents’ exposure to trauma and the health, wellbeing, and development of their children [21,24] adds to the urgency of developing and implementing effective strategies to support fathers of refugee background as they navigate life in countries of settlement. Consistent with the socioecological model, interventions at individual, family, and community levels are all crucial to promoting recovery from trauma for people of refugee background, acknowledging the ways in which risk and protective factors interact across these levels [41]. Our findings strongly support the need for services and providers working with fathers of refugee background to take a trauma-informed approach [41,42]. A trauma-informed approach to working with people of refugee background emphasises safety, including advocating for access to basic needs, reducing barriers to service access, and taking steps to optimise physical, emotional, and cultural safety in healthcare settings [41,43]. Restoring secure attachments through prioritising the development of positive, trusting working relationships with clinicians and facilitating opportunities for greater social connectedness is also an important aspect of supporting recovery from trauma [41]. Trauma-informed services take a strengths-based approach that supports clients of refugee background to restore a sense of identity and meaning [41]. The community-based setting and longitudinal nature of primary healthcare means these services are well placed to support fathers of refugee background with their recovery from traumatic experiences.

Despite the challenges they faced, fathers demonstrated a great deal of resilience, drawing on sources of strength from within themselves as individuals and from their social and community networks. Some fathers also accessed support from healthcare services. Fathers closely engaged with their children and derived a strong sense of meaning from parenting. While many were limited in the time they had available to themselves, several described personal coping mechanisms that they drew upon in challenging times. Close relationships with their partner and family members were important sources of strength. Findings reflect how fathers of refugee background experience protective factors across multiple environmental subsystems. Health and social care services aiming to improve fathers’ mental health should be mindful of the supports that fathers currently draw upon and consider ways to build upon or integrate these in interventions.

Most fathers in the study reported being in good health. However, as already noted, there is evidence from other Australian research showing higher rates of poor mental and physical health among fathers of refugee background compared with fathers not of refugee background [19]. Our findings highlight the ongoing impacts of trauma and multiple stressors experienced in the period after settlement. Notably, our study supports the findings of Riggs and colleagues [25], indicating that while fathers are actively involved in accompanying their wives and children to healthcare appointments, they are rarely asked about their own health or wellbeing. Compared to women, who may be more likely to seek healthcare for reproductive health or pregnancy-related needs, fathers of refugee background with young children may be less likely to seek healthcare for themselves. When fathers accompany their partners and children to healthcare appointments, this presents an opportunity for engagement and building relationships to support fathers’ health and wellbeing. Our findings also strongly support the need for health services and health practitioners to be mindful of how fathers of refugee background access, understand, and apply health information in order to overcome communication, health literacy, and language barriers.

### 4.1. Strengths and Limitations

To our knowledge, this is the first qualitative study to ask fathers of refugee background with young children living in Melbourne, Australia, about their experiences of risk and protective factors for their health and wellbeing. Strengths of the study include: the use of participatory methods to engage fathers in the study, conducting the research in fathers’ preferred language and environment, and the design of the study to provide participants with the option of participating individually or in a group. Nonetheless, several limitations apply. While the broad research question resulted in a broad overview of fathers’ experiences of risk and protective factors affecting their health and wellbeing, further research is needed to achieve a more in-depth understanding of a number of areas. In particular, understanding the ways in which experiences and needs evolve across different stages post-settlement and how services and policies should respond to these should be an area of further study. Our study did not explore the differences between the experiences of fathers whose children were born overseas compared to those whose children were born in Australia, who may have different experiences of accessing services. All fathers interviewed were living in a metropolitan area and there were no fathers in our sample who had been in Australia for less than 12 months. The needs of those living in rural and regional areas and those who are newly arrived may differ and should be explored in future research.

### 4.2. Implications

Our findings support the need for a holistic approach to improving the health of fathers of refugee background, informed by the ways in which fathers experience risk and protective factors across individual, family, community, and broader societal levels. Results indicate there is a need for improved support for fathers of refugee background across the settlement process, particularly with areas like securing employment and learning English. Social policies and community interventions in these areas should be mindful of the particular needs of fathers of young children, including the additional financial pressures and multiple demands that fathers may be managing. Pregnancy and early childhood healthcare settings provide multiple opportunities to engage and build relationships with fathers of refugee background. Our study supports the need for maternity and early childhood services to take a broader, family-inclusive approach. Professional development for healthcare providers working with refugee communities on how to engage and include fathers in these settings and develop culturally sensitive approaches that promote the health and wellbeing of fathers of refugee background would be beneficial. Findings strongly support the need for health and social care services to take a trauma-informed approach, underpinned by a strong awareness of the high rates of trauma exposure among refugee families and sensitivity to the needs that arise from this. Further research, including the evaluation of trauma-informed approaches to care, should continue to build an in-depth understanding of the health needs of fathers of refugee background and how to meet them.

## 5. Conclusions

This study provides insight into how fathers of refugee background settling in Australia experience risk and protective factors for their own health and wellbeing during their children’s early years. Our findings highlight that many fathers of refugee background have experienced significant trauma prior to arrival, and then face further major stressors as they navigate new systems and raise their children. Despite this, fathers demonstrate extraordinary resilience. The study clearly identifies the need for systems reforms to improve the way that health and social care services engage with and promote the health, wellbeing, and resilience of fathers of refugee background. There is an urgent need to implement and evaluate trauma-informed approaches to health and social care that recognise and build on the strengths of fathers during the critical phase of establishing their families in a new country. Improving the health and wellbeing of fathers of refugee background is critical to the achievement of health equity for all family members.

## Figures and Tables

**Figure 1 ijerph-19-06940-f001:**
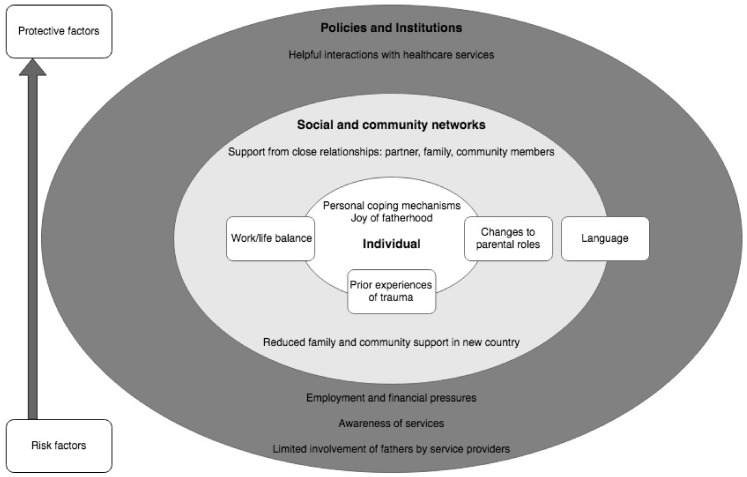
Socioecological map of identified themes.

## Data Availability

Data not available due to the conditions of ethical approval. Participants were assured that raw data would not be shared.

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
