# Peer review of "Risk and Protective Factors Experienced by Fathers of Refugee Background during the Early Years of Parenting: A Qualitative Study"

_ijerph, 2022, doi:10.3390/ijerph19116940_

Round 1

Reviewer 1 Report

The study focuses on fatherhood among specific population: refugees who are trying to adapt to new environment and daily challenges of this situation. The theme is clearly described, methodology is precisely presented. The distribution of topics drawing from the socioecological model applied is suitable, however, the model itself is standard, and it does not bring substantial advances for interpretation of otherwise exceptionally rich qualitative data. The authors, yet, overcome the limits of the model with thorough contextualization, deep-understanding of the complex situation fathers of refugee background are in and indication of nuanced entanglements of the web of relations between different subjects refugee fathers are encountering and their implications. The impact of the findings can first be seen in drawing attention of relevant services to this particular issue, which has not been systematically discussed yet, and then improving their support of fathers with refugee background. 

Reviewer 2 Report

It is an interesting paper presenting original research concentrating on how fathers of refugee background experience risk and protective factors for their health and well-being. The study is qualitative in nature and is limited to Melbourne metropolitan area, Australia. Semi-structured individual interviews and one focus group were organized. Data from 21 participants were collected. Thematic analysis and the socioecological model of health were used to extract meaningful information and interpret the outcomes. The strength of the study is the filling of the research gap, as fathers are underrepresented in studies on parents of refugee background. The authors identified personal, social and community network as well as policy and institutional level factors. Their description is supported by citations from the interviews/focus group. This exploratory study is a valuable contribution to the research on parents of refugee background. Findings have also a practical aspect for social and healthcare providers.

I have dome detailed remarks:

Line 112 and lines 155-156. Please clarify information about the age of participants. Line 112: “The participants were men of refugee background, aged 18 years and older”, lines 155-156: “The mean age of participants was 40 years (range 28-56 years; standard deviation 6.43)”. I understand that line 112 says about the possible age assumed by researchers (target group or population, potential participants) but it would be better to underline it.

Line 168: It is not clear how to interpret some elements in Figure 1. From bottom to top, there's a change from risk factors to protective factors illustrated by an arrow shown on the left side. Elements like work/life balance, changes to parental roles and language are just in the middle of the figure corresponding to the central point of this arrow. Does it mean that they are neutral, i.e. neither risk nor protective? Or are they not evaluated on this scale as they are enclosed in rectangles and not given as plain text like other factors? A comment on it in the description above the figure would be helpful to the reader.

Lines 438-439: “All fathers interviewed were living in metropolitan areas”. Is plural “areas” adequate? Or is it one metropolitan area (Melbourne) as it was described in the manuscript?

Reviewer 3 Report

I really enjoyed reading your article and have only a very few comments/suggestions for you. First, you employ passive voice throughout much of your piece, which left me a bit at sea in following who actually undertook many of the actions you describe. It may seem a small thing but your very fine paper would be much stronger if you undertook a careful edit to rid it of passive voice. Second, I noted you employed a professional transcriptionist which, of course, is fine. You did not say, however, that you had asked that service or individual to sign a confidentiality agreement? Nor, did you indicate whether you had cross-checked transcripts yourselves against recordings to ensure their accuracy?  Third, it is unusual in my experience to offer both verbal and/or written consent to participants. Can you indicate why you did so? I assume as I write that all whom you interviewed were literate and they they consented to documents prepared in an appropriate language? It would be helpful to clarify/provide a clear rationale on this point. Finally, your suggestion concerning employing trauma informed approaches at line 393 and following struck me as appropriate. I think it might be useful to include an additional paragraph or two to tease out how, more precisely, that construct should guide future work with this population concerning the stressors of interest.
